# High-Resolution Snow-Covered Area Mapping in Forested Mountain Ecosystems Using PlanetScope Imagery

Aji John [1,*], Anthony F. Cannistra [1], Kehan Yang [2,3], Amanda Tan [3], David Shean [2], Janneke Hille Ris Lambers [1,4] and Nicoleta Cristea [2,3]

1. Department of Biology, University of Washington, Seattle, WA 98195, USA; tonycan@uw.edu (A.F.C.); janneke.hillerislambers@usys.ethz.ch (J.H.R.L.)
2. Department of Civil and Environmental Engineering, University of Washington, Seattle, WA 98195, USA; kyang33@uw.edu (K.Y.); dshean@uw.edu (D.S.); cristn@uw.edu (N.C.)
3. eScience Institute, University of Washington, Seattle, WA 98195, USA; amandach@uw.edu
4. Plant Ecology, Institute of Integrative Biology, Department of Environmental Systems Science, ETH Zürich, 8092 Zürich, Switzerland
* Correspondence: ajijohn@uw.edu

**Abstract:** Improving high-resolution (meter-scale) mapping of snow-covered areas in complex and forested terrains is critical to understanding the responses of species and water systems to climate change. Commercial high-resolution imagery from Planet Labs, Inc. (Planet, San Francisco, CA, USA) can be used in environmental science, as it has both high spatial (0.7–3.0 m) and temporal (1–2 day) resolution. Deriving snow-covered areas from Planet imagery using traditional radiometric techniques have limitations due to the lack of a shortwave infrared band that is needed to fully exploit the difference in reflectance to discriminate between snow and clouds. However, recent work demonstrated that snow cover area (SCA) can be successfully mapped using only the PlanetScope 4-band (Red, Green, Blue and NIR) reflectance products and a machine learning (ML) approach based on convolutional neural networks (CNN). To evaluate how additional features improve the existing model performance, we: (1) build on previous work to augment a CNN model with additional input data including vegetation metrics (Normalized Difference Vegetation Index) and DEM-derived metrics (elevation, slope and aspect) to improve SCA mapping in forested and open terrain, (2) evaluate the model performance at two geographically diverse sites (Gunnison, Colorado, USA and Engadin, Switzerland), and (3) evaluate the model performance over different land-cover types. The best augmented model used the Normalized Difference Vegetation Index (NDVI) along with visible (red, green, and blue) and NIR bands, with an F-score of 0.89 (Gunnison) and 0.93 (Engadin) and was found to be 4% and 2% better than when using canopy height- and terrain-derived measures at Gunnison, respectively. The NDVI-based model improves not only upon the original band-only model's ability to detect snow in forests, but also across other various land-cover types (gaps and canopy edges). We examined the model's performance in forested areas using three forest canopy quantification metrics and found that augmented models can better identify snow in canopy edges and open areas but still underpredict snow cover under forest canopies. While the new features improve model performance over band-only options, the models still have challenges identifying the snow under trees in dense forests, with performance varying as a function of the geographic area. The improved high-resolution snow maps in forested environments can support studies involving climate change effects on mountain ecosystems and evaluations of hydrological impacts in snow-dominated river basins.

**Keywords:** PlanetScope; snow cover mapping; forests; snow; machine learning; convolutional neural networks

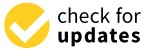



## 1. Introduction

Mountain ecosystems in the western US are very sensitive to changes in climate [1–3]. Snowpack is melting earlier [4], affecting the fragile alpine systems for, e.g., the peak wild-flower season that closely follows the snow disappearance day each year is shifting [5,6]. Unprecedented changes to mountain forests from fires and insect outbreaks [7] are also the result of changing climates, and forest managers are already opening gaps in areas of dense forest to improve ecosystem health and resilience [8]. Ecologists and conservation biologists study these systems using in situ monitoring, which is limited to point observations, precluding the understanding of the ongoing changes in areal extent and magnitude [9].

Changes to snow cover in particular are likely to influence mountain forests and meadows, because snow cover in the mountains dictates the growing degree days (accumulation of warmth). Thus, snow cover can affect the phenological advancement in species, causing the emergence of pollinators (insects etc.). Earlier snowmelt could lead to earlier flowering, which might cause asynchrony with pollinators or make them susceptible to frost events [10,11]. In a warming climate, early spring can also lead to spring runoff causing flooding and drier summers [12,13]. Understanding the heterogeneity of snow cover has implications that could inform biologists about the changes in ecosystems and water managers about the snow cover variability determined by seasonal snowmelt, which is a major component in hydrological water estimates [14].

The analysis of high-resolution (m-scale) lidar-derived snow depth datasets provides insights into the main factors controlling snow distribution and melt patterns, with important drivers shown to be elevation, followed by slope, aspect, and vegetation cover. With the lidar-derived snow depth datasets, one can observe snow in openings, in forest gaps, and under forest canopies, and how it varies in space and time at high spatial resolutions of 1–3 m [12,15]. Airborne lidar datasets are known to be reliable in their estimates of snow depths in forested areas through comparing airborne lidar snow depths with manual measurements of snow around the trees [12] and are generally found to agree within 5 cm at 0.5–5 m resolutions [16]. High-resolution snow-covered areas can be derived from the lidar-derived snow-depth data using a prescribed threshold [17]. In 2013, NASA started airborne snow depth collections in the Tuolumne watershed in California, USA, weekly during the ablation season under a program called the Airborne Snow Observatory (ASO [18]). The airborne lidar snow depth collections have since been expanded to other areas, but it is cost prohibitive to expand at a global level, and the collections remain limited.

Fortunately, high spatiotemporal resolution imagery is available via Planet Labs, Inc. (Planet) [19] which has the potential to transform the study of earth-processes through remote sensing. Planet operates a constellation of CubeSats—small satellites that are of 3U form factor (10 by 10 by 30 cm) that images the entire land surface area of the earth daily. The "PlanetScope" constellation is composed of roughly 130 satellites that operate in sun-synchronous orbit. The resulting product is orthorectified surface reflectance (SR) that is delivered at 12-bit resolution with 3–5 m resolution covering visible and near infrared bands [20].

Of course, detecting snow from remote-sensing imagery requires a model to translate bands to predictions of snow, and machine learning (ML)-based methods have proven to be extremely promising at detecting snow cover. The majority of studies have used multispectral providers (e.g., MODIS and Landsat-8) because of the band diversity. Commonly used methods are support vector machines (SVM) and random forests (RF), but deep learning-based methods are being applied increasingly [21]. Along with the multispectral bands, derived indices have also been used (for e.g., Normalized Difference Vegetative Index (NDVI) and Normalized Difference Snow Index (NDSI)), and yield snow cover area (SCA) estimates that have coarser spatial resolutions (e.g., 500 m) but finer temporal resolutions (e.g., daily) [22]. Studies that applied a suite of ML methods to a multispectral dataset combined with ancillary information such as topography found that the relevance of non-spectral attributes is of limited importance in increasing model performance [23,24]. Spatial resolution and accuracy tend to be issues in mountainous areas when using multispectral data (for e.g., MODIS)—-coarser data products can capture less of environmental

variation in very heterogeneous environments. Newer approaches have begun fusing multispectral satellite data with unmanned aerial vehicle (UAV)-based acquisitions or only using UAV-produced datasets to improve deficiencies over localized areas [25,26]. Another hybrid approach has also been proposed, where a convolutional neural network (CNN) is used to extract features and then a RF-based method is then used to estimate the snow-covered area [27]. Other explorations using satellite spaceborne synthetic aperture radar (SAR) data instead of multispectral have also shown promise in mapping snow-covered areas; SAR-based methods are not as prone to the presence of clouds, but SAR data have limitations in dense forests [28]. Furthermore, recent work by Cannistra et al., 2021 [29], has demonstrated that snow cover can be successfully mapped from PlanetScope data using a machine-learning approach based on a CNN using the 4-band (R, G, B and NIR) PlanetScope surface reflectance data as input. The study highlighted the viability of a CNN-based model trained on lidar-derived snow cover data despite limited radiometric bandwidth and band placement, but, similar to other satellite-derived data, the performance of the model is lower in forested areas.

Snow cover mapping in forested areas remains a complex challenge as it is driven by vegetation type, topography, and climate. The accumulation patterns of snow in forests vary as a function of distance from the canopy—under the canopy, to canopy edge, and in gaps [12,21]. Dense canopy contributes to shading caused by tall trees and the proximity of varied overstories moderates the interception of snow [30]. Furthermore, the directional variance around the trees has also been found to influence snow accumulation patterns [15]. The relative influence of those factors on snow depth varies as a function of location and the snow season [31]. For instance, Tennant et al., 2017 [32], showed that elevation explained most of the variability in snow depth (16–79%) in forested areas, but the aspect explained more variability (11–40%) in open areas. Cristea et al., 2017, also showed that terrain may matter more in the open than in vegetated areas. Snow–forest interactions also vary as a function of climate and radiation effects, with snow disappearing first in the open or under the forest as a function of local conditions [33]. These studies demonstrated that small-scale variability is being observed with lidar and identified the terrain and vegetation features controlling the spatial distribution of snow depth and, hence, snow cover. Based on these observations, we hypothesized that augmenting CNN-based models with terrain-derived and vegetation predictors is likely to improve predictive models in forested areas (e.g., [32,34]).

Therefore, in this study, we augment the Cannistra et al., 2021, CNN snow cover model by using additional predictors including vegetation structure (using lidar-derived canopy height from a canopy height model (CHM) and the Planetscope-derived Normalized Difference Vegetation Index (NDVI)), and the digital elevation model (DEM or elevation) and its derived attributes (i.e., slope, aspect and northness). It is important to note that the NDVI as a reliable predictor for canopy is supported by its sensitivity to vegetation response despite the availability of many other vegetative indices [35]. We then evaluate if these augmentations lead to improvement upon the original band-only model performance using the produced m-scale SCA from PlanetScope imagery. In our assessments we test two hypotheses: (1) terrain-derived predictors and vegetation information improve snow mapping accuracy in both forested areas (FA) and open areas (OA), and (2) terrain-derived predictors such as slope and aspect are more accurate over open areas than in forested areas where elevation is more important. We evaluate performance across forested areas, near the canopy edge, and in open areas (gaps) using a set of canopy classification metrics.

## 2. Study Areas

We used three study sites for our analysis: one site for the model training/validation, the Tuolumne River Basin in Sierra Nevada of California, USA (37.89°N, −119.25°W) and two sites for additional model evaluation: Gunnison/East River Basin in the Central Rocky Mountains of Colorado, USA (39.08°N, −107.14°W) and a site near Engadin in Switzerland (46.58°N, 10.03°W) (Figure 1). We selected these sites as they are part of Airborne Snow Observatory (ASO) monitoring [36] and had overlapping PlanetScope

imagery. Site characteristics are varied at the three sites; they are dissimilar in elevation, climate, and differing forest cover (sparse vs. dense) and climatic zones (maritime vs. continental). The Tuolumne basin has elevations ranging between 1500–3970 m and is in the central Sierra region of California; the area is a mix of evergreens and shrublands [36]. The Gunnison site is in southwest Colorado and has elevations ranging between 1387–4359 m [37]. The Engadin site is in the Grison region of Switzerland and ranges in elevation between 1700–2000 m; the site is primarily evergreen and composed of spruce and larch forest with homogeneous understory [12].

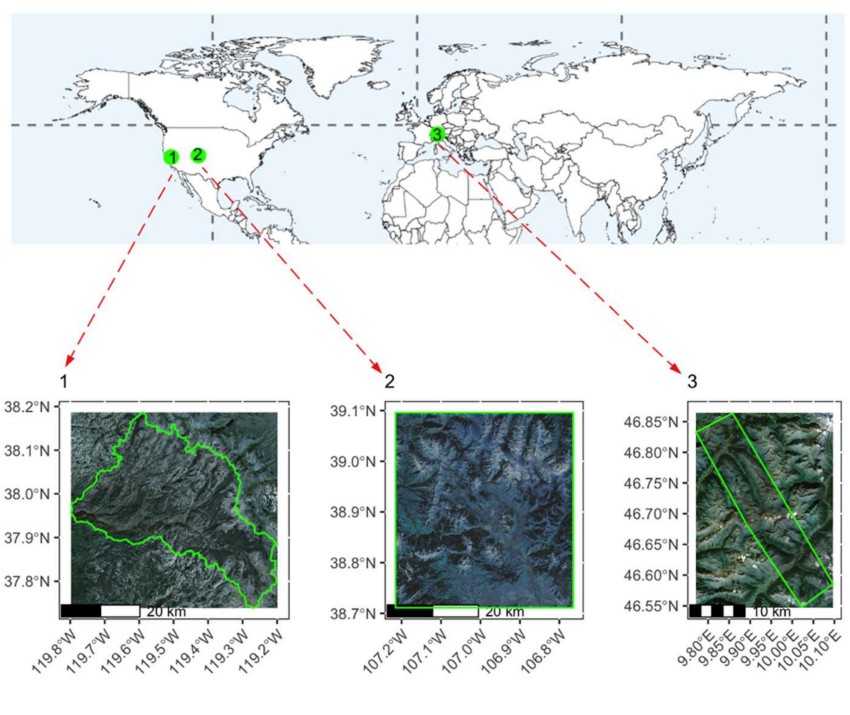

**Figure 1.** Study sites—(**1**) Upper Tuolumne (37.89°N, −119.25°W, California, USA), (**2**) Upper Gunnison (39.08°N, −107.14°W, Colorado, USA), and (**3**) a site near Engadin, Switzerland (46.58°N, 10.03°W, Switzerland).

## 3. Data and Methods

### 3.1. Data Products

We augmented the training datasets used in Cannistra et al., 2021, to include data from multiple sites for vegetation and terrain. The overlapping PlanetScope imagery used for training and validation is provided in Appendix A. The PlanetScope constellation includes multiple generations of CubeSats starting in 2014. Many of these satellites are in a sun-synchronous orbit with 4 to 8 band radiance products at approximately 3–4 m ground-sampling distance. Earlier generations (Dove Classic) offered 4-band products (RGB and NIR), while newer generations (Dove-R and SuperDove since 2019) provide 8-bands. In this study, we use the 4-band products, which enables longer time series analyses of the full PlanetScope archive.

As terrain features and vegetation predictors are likely to improve predictability in forested areas, we used lidar-derived datasets collected as part of the original NASA ASO campaign and additionally by the Airborne Snow Observatories, Inc. The lidar-derived digital elevation model (DEM), the canopy height model (CHM), and the snow depth product (true snow cover) were provided at 3 m resolution (Figure 2) by Airborne Snow Observatories Inc. (See Appendix A for the list of ASO product IDs used in this study). We followed similar methods outlined in Cannistra et al., 2021, using a snow depth threshold value to derive benchmark snow cover masks. We tested several thresholds (Section 4.3)

and determined that a threshold greater than or equal to 10 cm indicated the presence of snow in lidar-derived products [29]. More details on ASO data products and methodology can be found in Painter et al., 2016.

A coarser level DEM was also extracted for the study areas via data provided by the Shuttle Radar Topography Mission (SRTM), a product through National Aeronautics and Space Administration (NASA) which is a 1 arc-second (approximately 30 m) product (downloaded via https://dwtkns.com/srtm30m/, accessed on 1 April 2022). The models were primarily trained on the Tuolumne dataset, except for the extension model where the neighboring San Joaquin basin was used to reproduce the original model from Cannistra et al., 2021. All the models were then validated using data from the Gunnison River Basin, CO, USA (May 2018) and Engadin, Switzerland (May 2017) sites to evaluate model transferability.

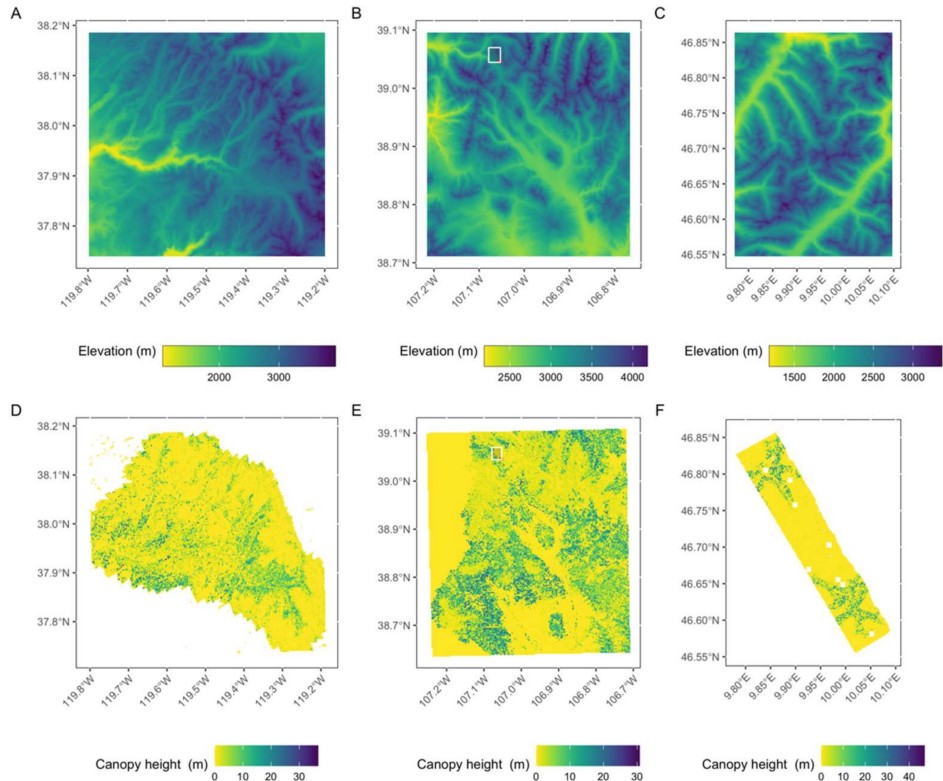

**Figure 2.** Elevation and canopy height for (**A**,**D**) Upper Tuolumne, USA, (**B**,**E**) Upper Gunnison, USA, and (**C**,**F**) Engadin, Switzerland.

### 3.2. Cyberinfrastructure

We extended the software and cyberinfrastructure developed by Cannistra et al., 2021 (Figure 3), who used a CNN-based method to identify snow-covered areas using only the 4-band PlanetScope Level-3 Analytic Surface Reflectance (SR) products. The Python-based implementation of the training procedure used PyTorch [38] and is a modified version of the "robosat.pink" software, an open-source set of command-line tools to enable machine learning with satellite imagery via "TernausNetV2"—a deep neural network that was modified after U-net for image segmentation [39]. Cannistra et al., 2021, modified the package to support the use of any N-band multispectral imagery data product and to allow the use of efficient cloud-based data storage and computation infrastructure. We similarly used highly scalable GPU-enabled "P2" instances from Amazon Web Services' (AWS) Elastic Compute Cloud (EC2) service to train the machine-learning models. Model training was performed with 50 epochs using AWS Sagemaker with default hyperparameters (learning rate of 0.000025 and batch size of 7). The code and all processing workflows are available on GitHub https://github.com/ajijohn/planet-snowcove-R (accessed on 1 April 2022).

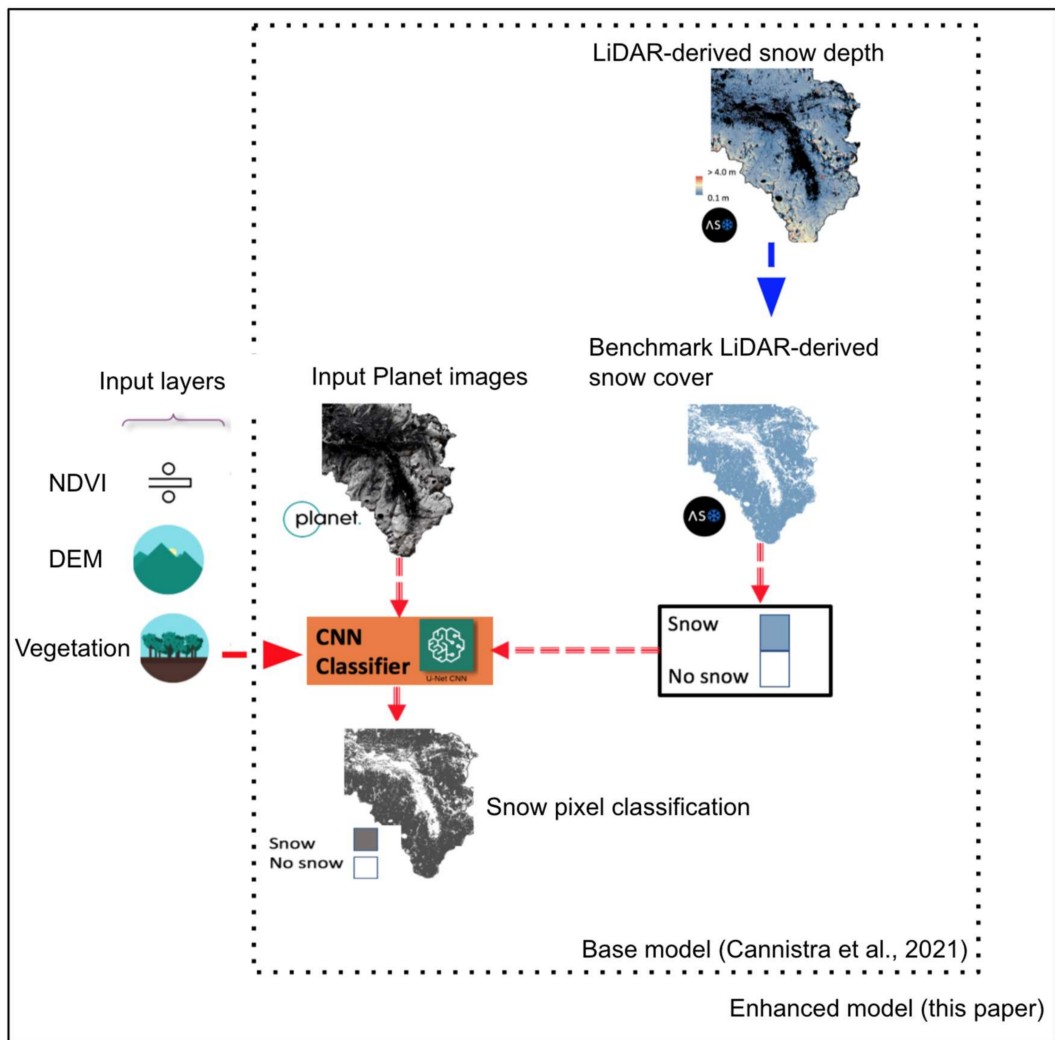

**Figure 3.** End-to-end workflow starting from pre-processing that involves downloading the Planet imagery and matching LiDAR derived snow cover data from ASO. In this study, existing bands are stacked along with DEM-based layers and vegetation proxies (Canopy height and NDVI). Model training is completed using the convolutional neural network model (TernausNetV2) on AWS with lidar-derived SCA as ground truth such as in the base model (Cannistra et al., 2021 [29]).

### 3.3. Model Augmentation

We call the Cannistra et al., 2021, CNN-based snow mapping model 'Model 0'. Model 0 was trained on Planet's 4-band PlanetScope data product that contains visible (Red, Green, Blue) and NIR (Near infrared); we refer to these sets of bands as the "BASE". To test hypotheses (that incorporating terrain-derived predictors and vegetation information into the model input data improves snow mapping in both forested and open areas), we enhanced Model 0 to include terrain and vegetation structure. These additional layers included DEM (elevation), slope, aspect, northness as (cosine(aspect) × slope) (e.g., see Tennant et al., 2017), NDVI, and CHM (vegetation height from the Canopy Height Model). To reiterate, we needed to improve the model performance and understand the model limitations in forested terrain. Due to that fact, we considered two types of vegetation representation in the model; through CHM and NDVI, and to draw on the influence of terrain, we used elevation (DEM) along with its derived features (slope, aspect, and northness). Slope, aspect, and northness were calculated using the methods described in Horn (1981). The Normalized Difference Vegetation Index (NDVI) is a normalized difference ratio of the NIR and red band with values ranging between −1 and +1 [40].

NDVI values around 0 generally signify non-vegetative landcover types like, water, snow, clouds, and rocks, and values closer to 1 signify vegetative growth [41]. We normalized CHM, DEM, and its derived attributes (slope, aspect, and northness) using "max–min" normalization whereby for a variable, a *base* is first established by taking the difference of its maximum and minimum, and then each of its values is subtracted by its minimum and divided by the *base*.

We tested individual input predictors alone and combinations of predictors to identify the best performing model configuration (Section 3.5 contains more details). The best performing model from these combinations was then used for further analysis. We term this Model 1. To test if terrain-derived predictors such as slope and aspect or vegetation structure are more accurate over open areas than in forested areas, we produced three grid-level canopy quantification metrics. We describe in detail the canopy quantification metrics in the following subsections. In total, six model configurations were evaluated for complete scene-based analysis and in-depth study based on the canopy metrics.

*3.4. Model Evaluation in Forested Areas*

To evaluate the performance of models over different forested terrains, we derived land-cover classifications using the canopy height model (Figure 4).

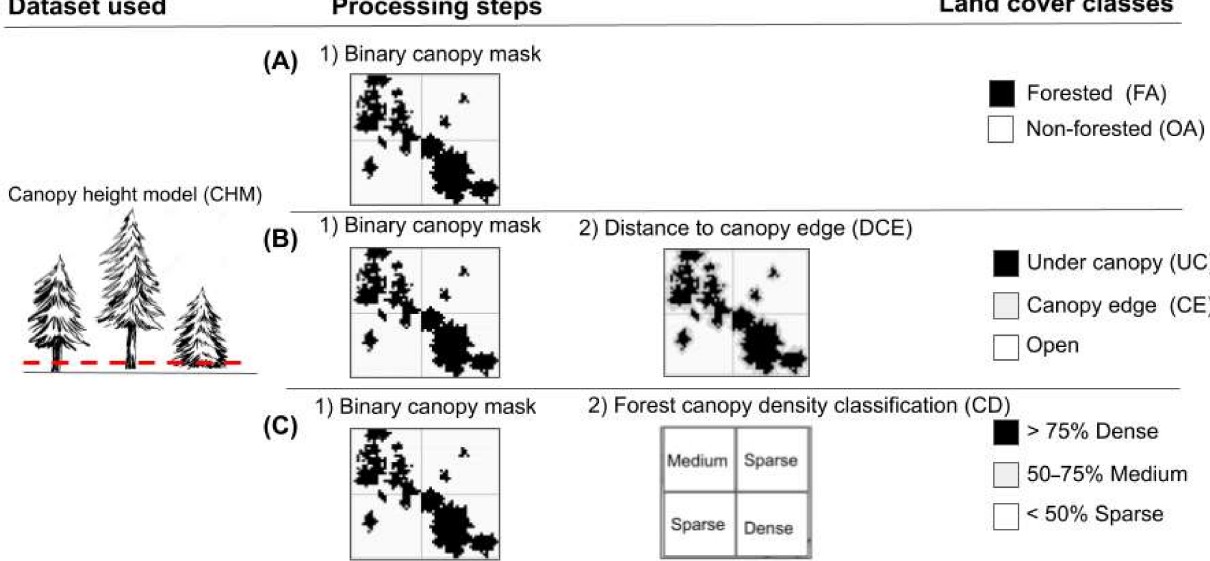

**Figure 4.** Framework for evaluating snow covered area mapping in forested terrain. (**A**) Classification based on canopy height; greater than 1 m (red line) is classified as forested area (FA), otherwise open area (OA). (**B**) Classification of land cover area: FA is composed of under canopy (UC) and canopy edge (CE), OA is outside CE; After Mazzotti et al., 2019 [12]. (**C**) Classification of land-cover area into forest density classes in which forest density is calculated in a larger focal pixel as a function of the vegetation presence (taller than 1 m) derived from the CHM. Forest density class is then assigned using the % of pixels. The coarse grid is 100 × 100 m.

For evaluation using binary maps derived from the canopy-height model, we used the 3 m canopy-height model (the ASO CHM) to derive a canopy-height-based binary grid (Figure 4A). We used a threshold of 1 m to separate forested areas from non-forested areas based on previous works [16] and to be consistent with previous model evaluations [29].

For evaluation using the distance to canopy edge (DCE), we use the Mazzotti et al., 2019, framework for further evaluation of the model's performance, particularly in the vicinity of the trees. For this evaluation we consider three classes as in Figure 4B: under canopy (UC), canopy edge (CE), and open. To delineate these individual classes, we use the distance-to-canopy-edge metric (DCE)—a CHM-derived metric used to study snow distribution in forested terrain [12]. Briefly, the algorithm starts with a CHM that is

thresholded at a prescribed height and a binary map is produced. A 3 × 3 running mean window is then applied to identify the canopy and near canopy area and then separate the area under the canopy (UC, black shade) and the canopy edge (CE, grey shade) using a threshold. We used a threshold of 30 m to separate the UC and CE areas. For comparison, Mazzotti et al., 2019, used a threshold in the range of 1 to 8 to categorize gaps, a range of −1 to 1 for edges, and a range of −1 to −3 to signify canopy clusters using a 1-m resolution CHM grid. The CHM used here to generate the DCE metrics in our study had a 3 m spatial resolution; therefore, we scaled the DCE accordingly to account for scale effects. We reclassified the DCE into three classes: values less than 0 were classified as UC—*under the canopy*, 0 to 30 as CE *near the canopy edge*, and greater than 30 as OA—*open areas*. UC refers to small or large clusters of canopies, CE refers to the edge of a canopy, and the Open category signifies gaps or open areas.

For evaluation using canopy density (CD), we used the 3 m canopy height model (CHM) product to produce a canopy density metric (Figure 4C). We first applied a threshold of canopy height greater than 1 m as a binary threshold and then used a moving average filter of 3 by 3 m to smooth it. The density was then calculated using percentage of grid cells with a canopy greater than 1 m within an area of about 100 × 100 m (Figure 4C, second processing step). Classifications were then made into *Sparse*, *Medium*, and *Dense* classes as a function of the percentage of vegetation pixels in the 100 × 100 m window (Figure 4C). More than 75% was considered *Dense*, between 50% and 75% was considered *Medium*, and anything less than 50% was labeled as *Sparse*. For example, a canopy pixel marked as *Dense* corresponds to 75% of pixels in the grid having vegetation of a height greater than 1 m.

*3.5. Model Performance Metrics*

We evaluated the performance of models over entire watersheds. In addition, we examined the models' performance in more detail in a subset of study areas and separately calculated the various land classes as described above. We used the following performance scores: Precision, Recall, F-score, Accuracy, and Balanced accuracy [42,43]. Model-derived SCA was compared with lidar-derived SCA snow cover (binary) at pixel level for all the metrics. "TP" ("true positives") is the number of pixels where snow is correctly classified as snow, and "TN" ("true negatives") is the number of pixels correctly classified as snow-free. "FP" and "FN" ("false positives" and "false negatives") represent the number of pixels that are incorrectly classified as snow and snow-free, respectively.

*Precision* (1) is the proportion of all true snow classifications (i.e., also true snow classifications in the lidar-derived SCA) of the model divided by all the snow classifications. *Precision* tells us how precise a model is, i.e., how good it is at detecting snow where there is in fact snow.

$$Precision = True\ Positives \div (\ True\ Positives + False\ Positives) \tag{1}$$

Similarly, *Recall* (2) is the proportion of all the true snow classifications by the model divided by all the predictions (correctly or incorrectly identified as snow) by the model. *Recall* tells us how good the model is at detecting all the snow pixels. Usually, a trade-off between *Precision* and *Recall* is preferred, but this can depend on the domain of which one is given more preference.

$$Recall = True\ Positives \div (True\ Positives + False\ Negatives) \tag{2}$$

The *F-Score* (3) is the harmonic mean of *Precision* and *Recall* that provides an overall performance that translates to the usefulness of the model. It ranges between 0 and 1 with higher values meaning a better predictive ability of the model.

$$F-score = 2 * Precision * Recall \div (Precision + Recall) \tag{3}$$

Another overall performance metric of the model we use here is *accuracy* (4), defined below.

$$Accuracy = (TP + TN) \div (TP + TN + FP + FN) \qquad (4)$$

*Balanced accuracy* (5) on the other hand, normalizes the "*TP*" and "*TN*" predictions when the corresponding binary classes are imbalanced (for example, "*TP*" are disproportionately larger than "*TN*"). Furthermore, the metrics were grouped by various canopy quantification metrics in a subset of study areas.

$$Balanced\ accuracy = (True\ positive\ rate + True\ negative\ rate) \div 2 \qquad (5)$$

*3.6. Role of DEM Resolution in DEM Based Models*

In our model development, we use high-resolution, m-scale lidar-derived DEM data, which is only available on select sites. In this step, as an alternative, we evaluate the use of a publicly available global DEM dataset such as SRTM in our models to assess the degree of degradation in performance of our models when using coarser-resolution digital-elevation-model (DEM) products. To test the influence of the DEM quality on the performance of our model, we upsampled the 30 m DEM (using SRTM) and extracted DEM-based attributes (slope, aspect, and northness) for training over the Tuolumne. We then retrained the models with the upsampled DEM data and used the model to validate the SCA at the Engadin and Gunnison sites.

## 4. Results

### 4.1. Overall Model Performance

We generated high-resolution SCA maps across three geographically diverse areas (Tuolumne Basin, CA, USA, Gunnison, CO, USA, and Engadin, Switzerland) for the various model configurations described above by training only on data from the Tuolumne basin. The SCA data for the Tuolumne basin were created only for the test dataset and was subsequently used for performance evaluation.

Table 1 shows the F-scores of all the predictor combinations. The results for Gunnison and Engadin indicate the model's performance over the entire study area. The least performing model was the one which used only 4-bands (visible and NIR), that we earlier defined as "BASE". The best performing model was the combination of base imagery with the NDVI and was able to perform better at Gunnison and Engadin. The addition of DEM (Elevation) resulted in better scores and derived features that included slope, aspect, and northness and resulted in lower scores than the NDVI-based model. The addition of CHM (canopy height) resulted in better scores than the BASE model and lower scores than the BASE + NDVI- or BASE + DEM-based models. The model that used slope, aspect, and northness was the least performant. In general, across the Engadin site, all candidate models performed better than the BASE model. The best performing model, BASE + NDVI, is referred to as Model 1 from here onwards. We limited the use of the BASEEXT model that included part of the San Joaquin basin, because we were able to obtain better overall performance in Gunnison and Engadin with a reduced training size and by only using existing bands. BASEEXT is the equivalent of Model 0 developed by Cannistra et al., 2021. Tables A2 and A3 contain detailed metrics. In addition to the predictor combinations in Table 1, BASE + DEM + CHM and BASE + DEM + NDVI were also evaluated but were dropped from further analysis because of low scores (F-scores of 0.65 and 0.31, respectively).

**Table 1.** F-scores of combinations of features (predictors). Metrics were calculated across 19 scenes (except for BASEEXT which was completed over 16 scenes) in Gunnison, and 2 scenes in Engadin. A scene is approximately 24 km by 8 km.

| Features | Tuolumne | Gunnison | Engadin |
|:---:|:---:|:---:|:---:|
| BASE [1] + DEM (Elevation) | 0.91 | 0.88 | 0.91 |
| BASE [1] + slope + aspect + northness | 0.67 | 0.73 | 0.90 |
| BASE [1] + DEM (Elevation) + slope + aspect + northness | 0.90 | 0.87 | 0.85 |
| BASE [1] + NDVI | **0.92** | **0.89** | **0.93** |
| BASE [1] + CHM (Canopy height) | 0.85 | 0.85 | 0.92 |
| BASE [1] | 0.32 | 0.85 | 0.90 |
| BASEEXT [1,2] | 0.76 | 0.88 | 0.92 |

[1] Refers to 4-bands (Visible and NIR) and tiles from Tuolumne basin only; [1,2] Refers to extended training that includes the San Joaquin basin.

Figure 5 shows the performance metrics for both the validation sites, Model 1 (BASE + NDVI) predictions were comparable to lidar-derived SCA at both the Gunnison and Engadin sites. The Gunnison River basin site had the following values: Accuracy of 0.89, Precision of 0.94, Recall of 0.85, F-score of 0.89, and Balanced Accuracy of 0.78 (Table A2 contains detailed metrics). Metrics across the entire Engadin site had the following values: Accuracy of 0.92, Precision of 0.93, Recall of 0.94, F-score of 0.94, and Balanced Accuracy of 0.91 (Tables A2 and A3 contain detailed metrics).

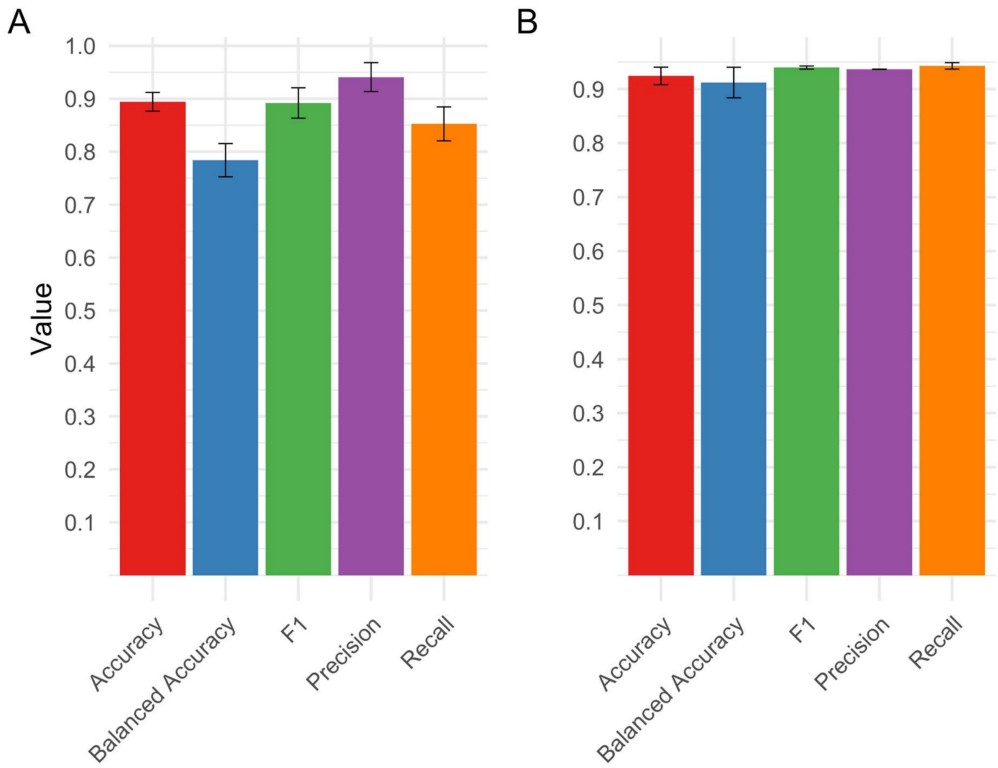

**Figure 5.** Overall performance (BASE + NDVI or Model 1) at Gunnison River and Engadin sites. (**A**) Mean metrics for Gunnison River basin, CO, USA; (**B**) mean metrics for Engadin, Switzerland. Bars show standard deviation.

Detailed metrics of the models provide insights into the intricacies of the model's performance (Figure 6). The NDVI-based model has the best-balanced accuracy, evident at Gunnison and Engadin sites (0.78 and 0.91). At the Gunnison site (Figure 6A), we note that even though BASE + NDVI is the best model, its recall is lower (0.85) compared to the slope + aspect + northness model (0.9); a similar pattern is found at the Engadin site as well. The DEM-derived model (DEM, slope, aspect, and northness) and DEM-only models consistently achieved the second-best performance at both sites. The BASE model is found to have the same precision as the best model (0.94) at the Engadin site.

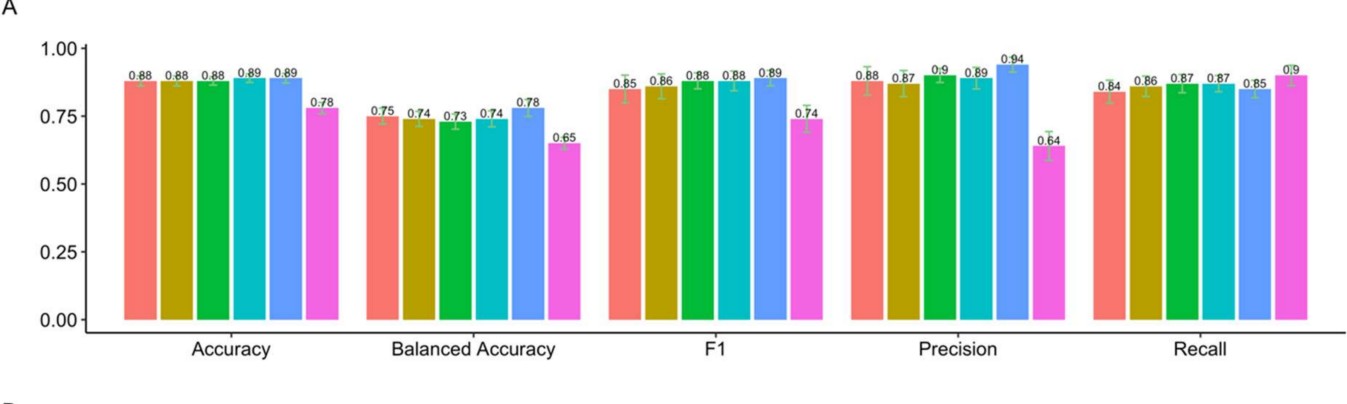

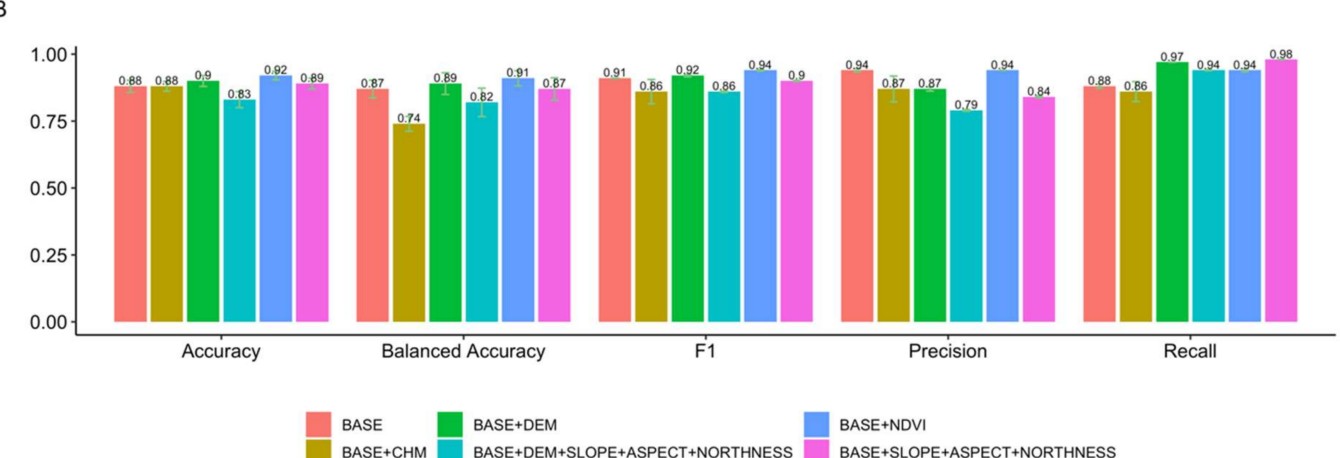

**Figure 6.** Detailed performance metrics of all the models at both the evaluation sites. (**A**) Complete metrics for Gunnison River basin, CO, USA. (**B**) Complete metrics for Engadin, Switzerland. Bars show standard deviation.

To examine the spatial distribution of snow in forested terrain more closely, we picked two areas within the larger Gunnison River domain that we further investigated (identified by white and red rectangles in Gunnison (Figure 2B,E)). Figure 7 depicts the evaluation of a larger area, which is mostly dense canopies, and Figure 8 shows the evaluation across a smaller area that is mostly open with sparse trees. The spatial distribution of the mapped snow of a sample site in a dense canopy and an open site at Gunnison show the nuances of the model's performance (Figures 7 and 8). Qualitatively, models generally underpredict snow in dense forest stands (lower left section of Figure 7), but DEM- and NDVI-based models perform comparatively well in comparison to rest of the models (Figure 7D,G). The lidar-derived SCA (Figure 7H) indicates that there is snow under dense canopies. Figure 7A shows the Planet image of the area, in which snow can be observed across the forested area. Therefore, the models, except for one, also predicted no snow, although the lidar dataset shows snow under the canopies. The CHM and BASE model overpredicts the snow in open (see along the diagonal and in the upper right section (Figure 7B,E)). The model with DEM and derived attributes (slope, aspect, and northness) overpredicts in open areas but is able

to obtain snow predictions correctly in ridges and valleys (see Figure 7C). The model Base + Slope + Aspect + Northness in general overpredicts snow in open areas and under the canopies (Figures 7F and 8F). The Base + NDVI model performs well across open areas (top right of Figures 7D and 8D). In the ensuing subsections, we describe these results further with detailed canopy quantification metrics.

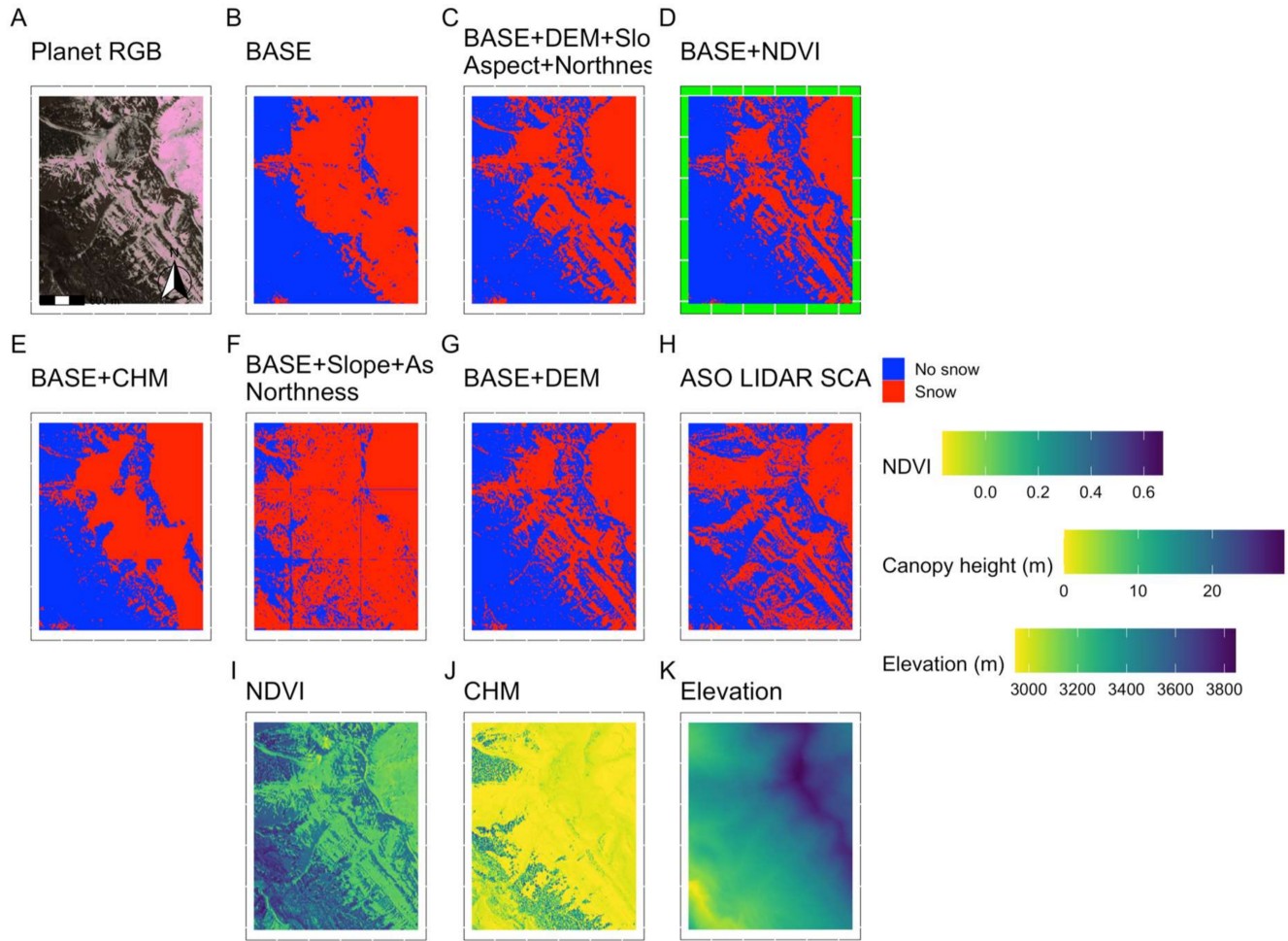

**Figure 7.** Performance of models in a sample area in Gunnison, which is in dense canopy and has canopy edges and open areas. (**A**) Planet RGB imagery showing the sample area in the Gunnison study site. (**B**) SCA predicted by the BASE model. (**C**) SCA predicted by the BASE + DEM (Elevation) + slope + aspect + northness model. (**D**) SCA predicted by the BASE + NDVI mode, the best performing model. (**E**) SCA predicted by the BASE + CHM model. (**F**) SCA predicted by the BASE + slope + aspect + northness. (**G**) SCA predicted by the BASE + DEM. (**H**) Lidar-derived SCA. Total study area of approximately 6.25 km$^2$. The highlighted model in green is the overall best model, Model 1. (**I**) Corresponding NDVI of the large study area derived from PlanetScope image. (**J**) Canopy height of the study area that shows the density in canopy cover. (**K**) Elevation of the study area. Note that the tiles were aggregated for this visualization and show some visible artifacts (border lines). The satellite imagery in panel (**A**) was downloaded from Planet Labs, Inc. All rights reserved.

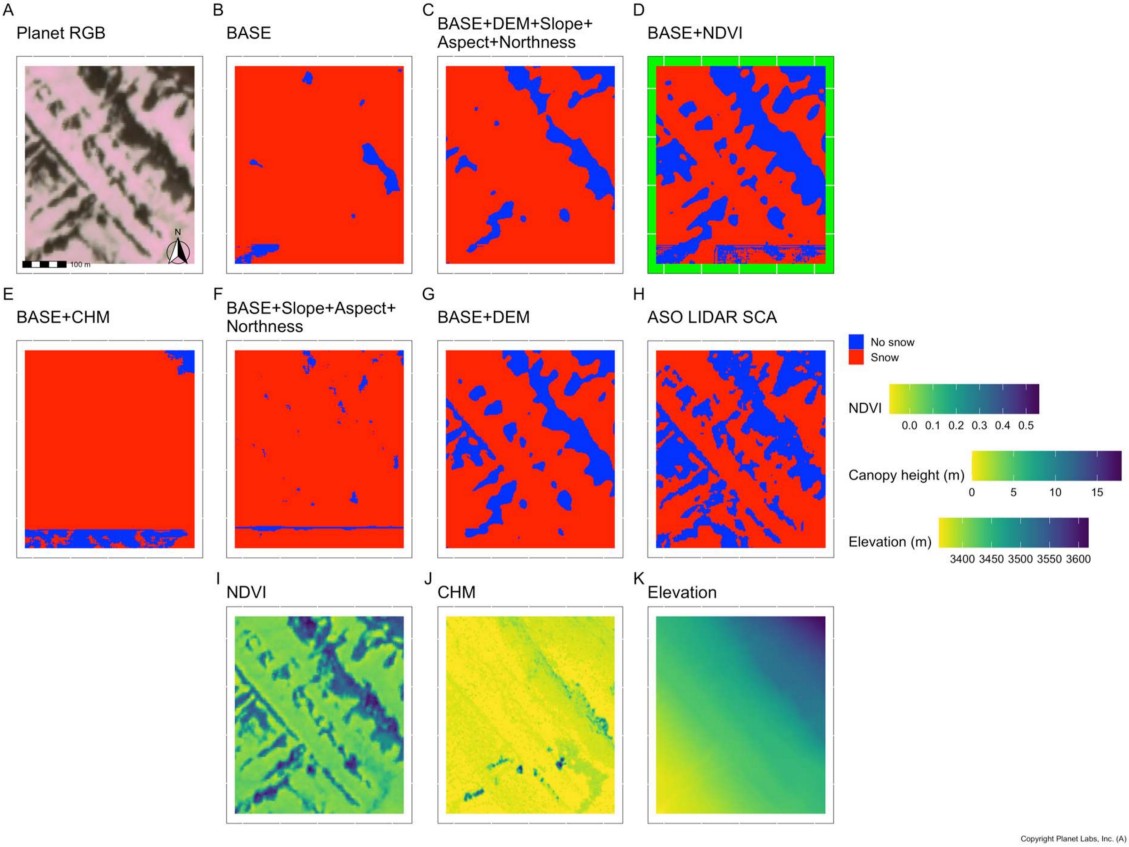

**Figure 8.** Performance of models in a sample area in Gunnison that is predominantly an open area. (**A**) Planet RGB imagery showing the sample area in the Gunnison study site. (**B**) SCA predicted by the BASE model. (**C**) SCA predicted by the BASE + DEM + slope + aspect + northness model. (**D**) SCA predicted by the BASE + NDVI model, the best performing model. (**E**) SCA predicted by the BASE + CHM model. (**F**) SCA predicted by the BASE + slope + aspect + northness. (**G**) SCA predicted by the BASE + DEM. (**H**) Lidar-derived SCA. Total sub study area of approximately 0.25 km$^2$. The highlighted model in green is the overall best model, Model 1. (**I**) Corresponding NDVI of the sample area derived from PlanetScope image. (**J**) Canopy height of the sample area that shows sparseness in canopy cover. (**K**) Elevation of the study area showing little variation. Note that the tiles were aggregated for this visualization and show some visible artifacts (border lines). The satellite imagery in panel (**A**) was downloaded from Planet Labs, Inc. All rights reserved.

### 4.2. Evaluation of SCA over Open and Forested Areas Using Canopy Metrics at Gunnison

At the Gunnison site, snow-mapping performance in the open and forested areas improved when DEM and vegetation-based predictors were used, and performance was better in open areas and at canopy edges than under the canopies. Table 2 shows the results using the various canopy metrics and land-cover classifications. In the open areas all the models but one, which included only slope, aspect, and northness, performed better than the BASE model. Using the CD metric, the vegetative models have better performance in sparse and mediumly forested areas compared to densely vegetated areas. In dense areas, the models with NDVI and DEM show similar performance. Using the DCE metric, we see better results in open and canopy edges across all the models except in the case of slope, aspect, and northness. The model with DEM and its derived attributes was found to be better under the canopy compared to all other models. Using a cut-off such as the CH metric, NDVI and DEM models were comparable in performance across open and forested areas.

**Table 2.** Performance metrics (F-score) in forest classes delineated using different forest canopy metrics within the Gunnison site across all the models. CD = canopy density, DCE = distance to canopy edge, CH = canopy height, CE—Canopy Edge, and UC—Under Canopy. The results were derived from 19 PlanetScope scenes for the Gunnison area.

| Canopy Metrics | CD | | | | DCE | | CH | |
|---|---|---|---|---|---|---|---|---|
| Model | Sparse | Medium | Dense | Open | CE | UC | OA | FA |
| BASE + DEM | 0.89 | **0.88** | **0.86** | 0.84 | 0.89 | 0.86 | 0.89 | 0.87 |
| BASE [1] + slope + aspect + northness | 0.69 | 0.74 | 0.79 | 0.67 | 0.69 | 0.78 | 0.70 | 0.74 |
| BASE [1] + DEM (Elevation) + slope + aspect + northness | 0.88 | 0.87 | 0.85 | 0.86 | 0.87 | **0.92** | 0.88 | 0.87 |
| BASE [1] + NDVI | **0.90** | **0.88** | **0.86** | **0.89** | **0.90** | 0.87 | **0.90** | **0.88** |
| BASE [1] + CHM (Canopy height) | 0.85 | 0.86 | 0.85 | 0.85 | 0.86 | 0.85 | 0.86 | 0.85 |
| BASE [1] | 0.85 | 0.85 | 0.84 | 0.82 | 0.85 | 0.85 | 0.84 | 0.86 |

[1] Refers to Visible + NIR bands.

### 4.3. Evaluation of Lidar-Derived SCA Threshold

The threshold that is used to produce a binary snow-covered area dataset from lidar-derived snow depth measurements might affect the performance, because snow depth is sensitive to the location of canopy edges. To assess the impact of this threshold on performance, we evaluated the change in the models' performance after varying this threshold. The threshold used to derive the lidar-derived SCA was varied across a range of thresholds at the Engadin site. We varied the threshold from 3 cm to 20 cm and recalculated the performance scores. The evaluation was carried out using the DEM-derived model (DEM, slope, aspect, and northness). The performance slightly decreased when the threshold was lower than 8 cm and above 10 cm (Figure 9).

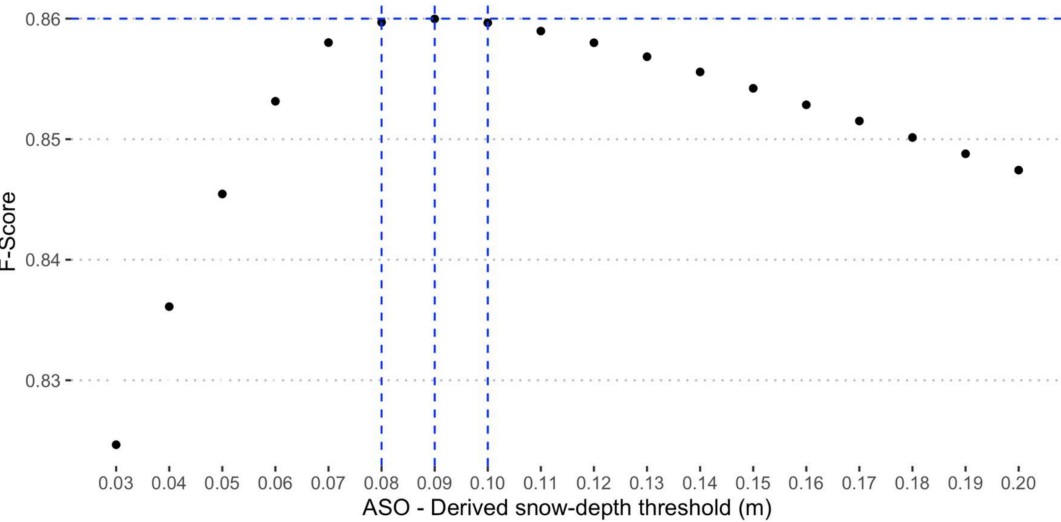

**Figure 9.** Model performance of DEM with derived attributes model when the lidar-derived threshold for snow cover was varied from 3 cm to 20 cm.

### 4.4. Sensitivity of the DEM Resolution for Training

Model performance at both the evaluation sites was comparable when the 30 m SRTM based DEM was used in the model training. The model using DEM with derived attributes (DEM, slope, aspect, and northness) was chosen for this assessment as it was the ideal model to evaluate the nuances of DEM resolution. At the Engadin site, the models trained using 3 m and 30 m DEM data had comparable performances, but across the Gunnison

site the performance was different across these two model types. In particular, the model trained using the 3 m DEM data had better recall and accuracy. At the Gunnison site, the SRTM-based model had lower precision than the 3 m model in the Gunnison evaluation, but the F-scores were nearly the same (Figure 10). The SRTM-based model was calculated over 18 scenes versus 19 scenes for the 3 m model, this was because the coarser SRTM model failed to distinguish the snow-vs.-no snow in one of the scenes at Gunnison.

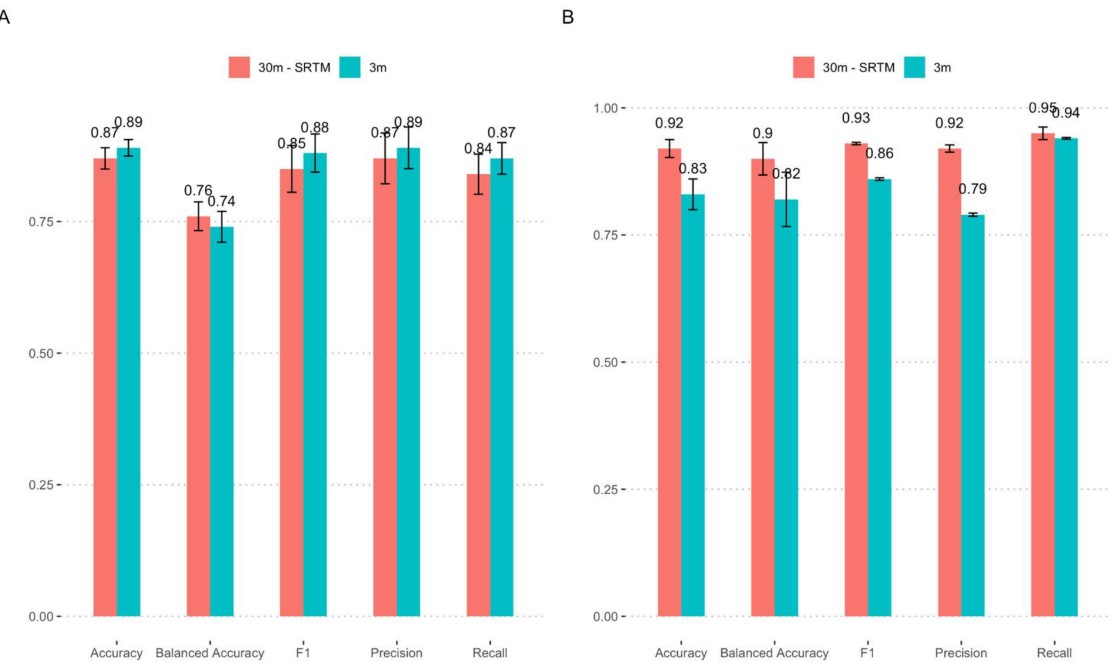

**Figure 10.** Model performance of DEM with derived attributes (DEM, slope, aspect and northness) model when 3 m was evaluated against a 30 m DEM (via SRTM). (**A**) Performance metrics over the Gunnison site. (**B**) Performance metrics over Engadin site. Bars show standard deviation.

## 5. Discussion

Overall, our results suggested that combining Planet with other landscape- or satellite-derived landscape features (especially elevation and vegetation) will improve predictions of snow cover, especially under dense canopies. However, modeling approaches, site locations, and data availability added complexities to predictions, implying that higher-quality remote-sensing products and further methodological improvements could further improve snow-cover predictions from Planet data. In our study, we evaluated the addition of new features to an existing CNN-based model originally based only upon unaltered inputs of Planetscope band data to map snow-covered areas from Planetscope imagery at m-scale spatial resolution. These additional features included a vegetative index (NDVI), canopy height, DEM, and topographic features (slope, aspect, and northness, the variables derived from the DEM).

Our hypothesis that terrain and vegetation structure improve the mapping of snow in forested areas was substantiated by the more accurate predictions of models using NDVI (proxy of canopy greenness) rather than terrain-based features (DEM, slope, aspect, and northness). The NDVI-based model was an overall best-performing model, and the DEM-derived model was better for pixels right under trees. We disproved our second hypothesis (where we suggested that the addition of slope, aspect, and northness features would improve performance more in forested areas than in open areas), with our finding that models including these factors overestimated snow in open areas contrary to our expectation.

### 5.1. NDVI Model (Preferred) Performance

We see in general that all the models perform better with band-based models (models which only use the spectral bands, which includes NDVI as well) still outperforming DEM- and canopy-height-based models. Given that the F-scores across all our predictor combinations are in near proximity (except for the model with slope, aspect, and northness), all of them are suitable candidate models. The band-based models capture the dynamics of snow cover, but the derived vegetation index (NDVI) is more precise (Precision of 0.88 vs. 0.94). Furthermore, the NDVI-based model has the highest balanced accuracy (0.78), a metric valued in case of imbalanced classes. The DEM-based models (except for slope-, aspect-, and northness-based model) have similar performances (both have F-scores of 0.88), suggesting that they would have similar predictive abilities, but that the DEM only model might be a more parsimonious choice. Canopy-based models are comparable to "BASE" (F-score of 0.86 and 0.85), suggesting that adding canopy height on top of the spectral bands does not drastically improve the model performance. The slope, aspect and northness model is shown to be less precise and less accurate (F-score of 0.65), but it does show high Recall. This is likely because of a high proportion of true positives (pixels correctly classified as no snow) compared to a significantly smaller number false negatives (pixels incorrectly marked as snow) but a higher number of false positives (pixels incorrectly marked as no snow).

We show that multispectral bands alone are sufficient in mapping snow-covered areas in forested areas. It is well known that band indices work well for deriving SCA, for example, MODIS-based snow cover product uses a combination of NDVI and NDSI to determine snow cover [44] and also more recent CNN-based models have found spectral bands to be sufficient at determining snow cover [23,29]. The NDVI-based model (Model 1) was the best model (F-score of 0.89) in our study. In this case, the model is likely associating the range of NDVI with presence or absence of snow, i.e., the higher the value of NDVI (closer to 1 for trees) and lower the value (close to −1) is getting associated as no snow (Figure 7E). However, this result is contrary to some other studies where NDVI was found to be limiting as a predictor [22,23]; however, this could be because of the coarser resolution of the product [45] that was used (MODIS has a 500 m multispectral resolution vs. the 3–5 m multispectral resolution of PlanetScope). Several studies using ML affirm the importance of multispectral bands in determining snow cover regardless of spatial resolution [22–24,44]. We find that other models were also relatively good, F-score varied by 15% across the remaining models; the elevation-based models being the second best, elevation is an important driver to the presence of snow and hence is better at delineating snow cover. NDVI is calculated using the existing bands, so no additional datasets are required.

### 5.2. Effect of DEM (Elevation) and Its Derived Attributes

Models using the DEM-derived attributes were also better at classifying snow in forested and open areas. This is consistent with other studies where contributions of northness, slope, and aspect have been shown to influence the presence of snow [24,30,32,46]. We found, in general, that the DEM-based models show comparable performance across different geographic areas (Figure 6A,B). Although they still underestimate snow in forest understories, they have better performance than the NDVI and CHM-based models. At Gunnison, the performance of the DEM-derived model (especially the one with slope, aspect, and northness) is better (F-score of 0.92 in Gunnison site using DCE) under canopies than rest of the models, but the same model is comparable to rest of the models in Engadin (to note, we had less snow on the ground, see Appendix B).

#### 5.2.1. Effects of the DEM Resolution on the Training Performance

We show the accuracy of SCAs generated using PlanetScope imagery in forested complex terrain is also subject to DEM resolution. The use of the fine resolution DEM (3 m) is found to improve the detection of snow (Figures 7 and 8) over the 30 m resolution DEM, the 3 m model provides us with an accuracy of 0.85 vs. 0.78 from the SRTM-based

model (Figure 10). However, we posit that the performance achieved in this study using coarser-level DEM might be reproducible in other geographic areas, because of the ready availability of coarser-resolution DEMs with global coverage [47].

### 5.2.2. Effects of the DEM Resolution on the Prediction Performance

We find that the coarser DEM was a reasonable replacement for the 3 m DEM. Specifically, the F-scores were comparable whether models incorporated the 3 m or the coarser DEM as the input (Figure 10). Over the Gunnison site ($n = 19$), the accuracy score with the 3 m DEM model with derived attributes as its input was 0.89 in comparison to a similar model that was trained using a coarser DEM, which gave a score of 0.87—a loss in accuracy of 2%. Moreover, with the coarser DEM, the model was not able to distinguish snow in 1 out of the 19 scenes. The difference in accuracy is not substantially different, but it does highlight the limitations in the use of coarser DEM when using DEM-based models. However, we deem that using DEM features from existing publicly available DEMs such as SRTM will ensure reliable model performance at other sites. Moreover, fine-resolution DEMs are not available everywhere, so the promise of a coarser DEM with a slight decrease in accuracy suggests that the model is theoretically applicable in many more geographic areas than it would be if only fine-resolution DEMs were used.

### 5.3. Applicability of Explored Models

Models can represent the SCA across forested terrain, but their skill in doing so varies as a function of forest-cover density. We note that the NDVI-based model can map the snow in open areas, gaps, and areas with sparse trees. The use of canopy quantification provided insights into model performance across canopy classes. Generally, models (including the best performing NDVI based model) performed better at canopy edges and in open areas than in the under-canopy areas. Models also showed significantly better performance in the under-canopy areas at the Engadin site than Gunnison. The F-scores across Gunnison ranged between 0.78–0.90 and were 6% higher than when using the BASE model.

The use of different canopy classes allowed us to benchmark model performance as a function of land-cover classifications and identify the critical thresholds at which the models succeed or fail. The addition of NDVI improves the performance (F-score of 0.87) of mapping snow cover in forested areas (Under Canopy metric in Table 2) and is better than CHM (Canopy height)- and DEM (Elevation)-based models in these areas [44,48], but the inclusion of DEM with slope, aspect, and northness is far more accurate (F-score of 0.92). The use of DEM derived attributes improves the prediction under the canopies where optical methods clearly have disadvantages.

We also note model improvements in different land-cover classes—open, sparse medium, and forested areas, with the performance varying by geographical area by 25% in F-score across the Gunnison site. Snow-covered tall vegetation would have a relatively higher albedo than snow-free shorter vegetation (<1 m) that has lower albedo; hence, optical collections such as PlanetScope are able to detect snow in low vegetative and open areas.

We document geographic differences in snow predictability in all our model evaluations. The F-score performance metric varies between 0.73 and 0.93. The snow characteristics at the two test sites inherently differ because of geomorphic and climatic differences. The characteristics of snowmelt dynamics are complex, as the effects of physiographic features play vital roles in snow accumulation and melt [30,49,50]. This is particularly relevant to our study which used training data from the snow ablation period, where snow distribution is most heterogeneous. We also caution that this might limit the model's transferability in a different geographical area (e.g., the Tibetan plateau [51]). A multi-site composite training might alleviate this issue and could be explored as a next step [23].

The vertical accuracy used in thresholding snow-depth was found to be reliable. The threshold value used for lidar-derived snow-classification played a minimal role in the differences we observed in the model's performance (Figure 9). We find that the performance is similar for thresholds from 8 cm to 10 cm and is lower for below 8 cm and

above 10 cm. This also suggests that any threshold between 8 and 10 cm should suffice in model development, a finding echoed in other works as well [18,29].

### 5.4. Model Feature Selection and Training Volume

Adding more features increased the training time proportionally, for example, a BASE model took an average of 4 h compared to a DEM-derived model (DEM along with slope, aspect, and northness) took five times longer. In machine-learning models, a simple model is preferred, as it prevents overfitting [52,53]; increasing the number of dimensions in the model input makes it more likely that the model captures both real and random effects.

### 5.5. Limitations

Firstly, we chose a limited set of predictors based on our physical understanding of snow dynamics in the system, and we believe it is likely that the models we ultimately selected would also perform well at predicting snow under trees in other snowy and forested systems. However, we suggest that those wishing to build similar predictive models in their own system consider other predictive variables to optimize their model to their local circumstances.

Secondly, the NDVI model captures the terrain dynamics as shown by the canopy quantifications but still misses snow in dense canopies. This is likely limited because of correlated reflectance in PlanetScope bands [54], variable radiometric quality, and the general difficulty in capturing the reflectance of snow via optical-methods in forest understories [48]. We speculate similar performance when other fractional band measures (e.g., the use of Green index that is different from the NIR and Green bands that are linked to forest canopy [55]) are considered, again, because of narrower bands by PlanetScope and the high degree of correlation between the bands [56]. We expect that the improved radiometric resolution and addition of new bands (such as the red edge) will help improve the predictions of snow in forest understories in the near future. Moreover, in general, snow under a forest canopy is challenging to observe via optical methods because of forest canopy cover and the resulting low signal-to-noise ratio [57,58]. Additional availability of bands (e.g., shortwave infrared (SWIR)) would also enable use of the Normalized Difference Snow Index (NDSI) that might better the predictability of the model in forests [23,59] and further gives the ability to mimic MODIS-type explorations that utilize broader band availability [44,48].

## 6. Conclusions

We used predictors such as vegetation structure (using canopy height and NDVI), digital elevation models (DEM), and DEM-derived attributes to produce 3 m snow-covered area from PlanetScope imagery to evaluate improvements in two representative important river basins, Tuolumne and Gunnison (in USA), and at Engadin (Switzerland). Overall, we find that the inclusion of NDVI into the model increases the model transferability more significantly than DEM and DEM-derived attributes such as slope, aspect, and northness. Our best model that used NDVI along with visible (red, green, and blue) and NIR bands captures the influence of vegetation on snow distribution. Specifically, the use of vegetation proxies (NDVI and canopy height) and terrain-derived measures was found to improve the accuracy of detecting snow in forested areas. The use of slope, aspect, and northness improves the ability of predicting snow in forest understories. The best model with an F-score of 0.89 (Gunnison) and 0.93 (Engadin) was found to be 4% and 2% better than when using canopy height and terrain derived measures at Gunnison, respectively. The NDVI-based model results in the best snow-identification performance in both forested and open areas compared to other models. Furthermore, adding only DEM and its derived attributes was also found to be transferable in test areas. The use of slope, aspect, and northness was found to overpredict snow in open areas. Even though optical methods are known to have shortcomings in observing snow in dense forest understories, our model's improvements, along with the detailed canopy-based evaluation metrics such as those presented here, can be used to improve model performance regarding various types of

forest feature (i.e., gaps, canopy edges, and dense overgrowth). Climate change projections show hydrologic changes to many mountainous basins; the approaches used in our study could be beneficial in mitigation efforts regarding climate change uncertainties. Improving snow-covered area identification in forested areas could improve hydrologic modeling accuracy and help to estimate late-season snowpack distribution. Our model holds promise, as it can better predict snow in forested areas that is in sync with captured imagery.

**Author Contributions:** Conceptualization, A.J., A.F.C. and N.C.; methodology, A.F.C. and A.J.; software, A.J. and A.F.C.; validation, A.J. and N.C.; formal analysis, A.J.; investigation, A.J.; resources, N.C.; data curation, A.J. and N.C.; writing—original draft preparation, A.J.; writing—review and editing, A.J., A.F.C., K.Y., A.T., J.H.R.L., D.S. and N.C.; visualization, A.J.; supervision, N.C.; project administration, N.C.; funding acquisition, N.C. All authors have contributed to manuscript review and editing. All authors have read and agreed to the published version of the manuscript.

**Funding:** This research was funded by the Data Science Environments project award from the Gordon and Betty Moore Foundation (Award #2013-10-29) and the Alfred P. Sloan Foundation (Award #3835) to the University of Washington eScience Institute, NASA (Award #80NSSC21K1151), NSF EAR—1947875 and NSF OAC—2117834.

**Acknowledgments:** We acknowledge the support from the University of Washington eScience Institute, UW Biology and UW Civil Engineering. We would also like to thank Kat Bormann from ASO, Inc. for providing data, and Giulia Mazzotti and Clare Webster for helpful discussions. Comments from three anonymous reviewers greatly improved the manuscript.

**Conflicts of Interest:** The authors declare no conflict of interest. The funders had no role in the design of the study; in the collection, analyses, or interpretation of data; in the writing of the manuscript, or in the decision to publish the results.

## Appendix A

**Table A1.** ASO collection IDs for the sites.

| Site | ASO ID [1] |
|---|---|
| Tuolumne, CA, USA | ASO_3M_SD_USCATE_20180528 |
| Gunnison River, CO, USA | ASO_3M_SD_USCOGE_20180524 |
| Engadin, Switzerland | EUCHDB20170517_SUPER |

[1] 3 m gridded product.

**Table A2.** Performance metrics in Gunnison River Basin, CO, USA [1].

| Metrics | BASE + NDVI | BASE | BASEEXT [2] | BASE + CHM | BASE + DEM | BASE + DEM (Elevation) + Slope + Aspect + Northness | BASE + Slope + Aspect + Northness |
|---|---|---|---|---|---|---|---|
| Accuracy | **0.89 (0.01)** | 0.88 (0.01) | 0.88 (0.01) | 0.88 (0.01) | 0.88 (0.01) | 0.89 (0.01) | 0.78 (0.02) |
| Precision | **0.94 (0.02)** | 0.88 (0.88) | 0.89 (0.02) | 0.87 (0.04) | 0.87 (0.02) | 0.89 (0.03) | 0.64 (0.02) |
| Recall | **0.85 (0.03)** | 0.84 (0.04) | 0.87 (0.02) | 0.86 (0.03) | 0.86 (0.03) | 0.87 (0.02) | 0.90 (0.03) |
| F-score | **0.89 (0.02)** | 0.85 (0.05) | 0.88 (0.02) | 0.85 (0.04) | 0.88 (0.02) | 0.88 (0.03) | 0.73 (0.04) |
| Balanced Accuracy | **0.78 (0.03)** | 0.75 (0.02) | 0.73 (0.02) | 0.74 (0.02) | 0.73 (0.02) | 0.74 (0.02) | 0.65 (0.02) |

[1] All the scenes in the study area (*n* = 19); scores represent mean values, and values in brackets are standard deviations. [2] Scores are over 16 scenes as the model was not able to predict snow over 3 scenes.

**Table A3.** Performance metrics in Engadin, Switzerland [1].

| Metrics | BASE + NDVI | BASE | BASEEXT | BASE + CHM | BASE + DEM | BASE + DEM (Elevation) + Slope + Aspect + Northness | BASE + Slope + Aspect + Northness |
|---|---|---|---|---|---|---|---|
| Accuracy | **0.92 (0.01)** | 0.87 (0.02) | 0.90 (0.005) | 0.90 (0.02) | 0.90 (0.02) | 0.83 (0.03) | 0.88 (0.02) |
| Precision | **0.93 (0.0009)** | 0.93 (0.006) | 0.90 (0.001) | 0.91 (0.003) | 0.86 (0.007) | 0.79 (0.003) | 0.83 (0.003) |
| Recall | **0.94 (0.005)** | 0.87 (0.007) | 0.93 (0.017) | 0.93 (0.013) | 0.97 (0.0001) | 0.93 (0.001) | 0.98 (0.0003) |
| F-score | **0.93 (0.002)** | 0.90 (0.001) | 0.92 (0.007) | 0.92 (0.004) | 0.91 (0.004) | 0.85 (0.002) | 0.90 (00.001) |
| Balanced Accuracy | **0.91 (0.02)** | 0.86 (0.03) | 0.88 (0.02) | 0.89 (0.03) | 0.88 (0.04) | 0.81 (0.05) | 0.87 (0.04) |

[1] All the scenes in the study area (*n* = 2); scores represent mean values, and values in brackets are standard deviations.

**Table A4.** 19 Planet Scene IDs (Dove Classic constellation) for Gunnison River Basin, CO, USA.

| PlanetScope Scene ID |
|---|
| 20180524_172142_103d_3B |
| 20180524_172143_103d_3B |
| 20180524_172144_103d_3B |
| 20180524_172145_103d_3B |
| 20180524_172146_103d_3B |
| 20180524_172147_103d_3B |
| 20180524_172148_103d_3B |
| 20180524_172326_0f51_3B |
| 20180524_172327_0f51_3B |
| 20180524_172329_0f51_3B |
| 20180524_172330_0f51_3B |
| 20180524_172331_0f51_3B |
| 20180524_172332_0f51_3B |
| 20180524_172333_0f51_3B |
| 20180524_172632_0f2d_3B |
| 20180524_172633_0f2d_3B |
| 20180524_172634_0f2d_3B |
| 20180524_172635_0f2d_3B |
| 20180524_172637_0f2d_3B |

**Table A5.** 2 Planet Scene IDs (Dove Classic constellation) for Engadin, Switzerland.

| PlanetScope Scene ID |
|---|
| 20170516_092514_1028_3B |
| 20170516_092515_1028_3B |

**Table A6.** 12 Planet Scene IDs (Dove Classic constellation) for Tuolumne Basin, CA, USA.

| PlanetScope Scene ID |
| --- |
| 20180528_180846_1002 |
| 20180528_180847_1002 |
| 20180528_181108_1025 |
| 20180528_181109_1025 |
| 20180528_181110_1025 |
| 20180528_181111_1025 |
| 20180528_181112_1025 |
| 20180528_181113_1025 |
| 20180528_181319_1005 |
| 20180528_181320_1005 |
| 20180528_181322_1005 |
| 20180528_181323_1005 |

## Appendix B

**Table A7.** Performance metrics (F-score) in forest classes using different forest canopy metrics quantification at the Swiss site across all the models. CD = canopy density, DCE = distance to canopy edge, CH = canopy height, CE—Canopy Edge, UC—Under Canopy. Note that DCE metrics cover only one scene. The results were again evaluated using two scenes for the Engadin area. In open areas, NDVI- and CHM-based models perform better compared to DEM-derived models. DEM with all the topographic attributes is the least performing model. Using the CD metric, the vegetative models have better performance than DEM and BASE models in sparse areas. In medium and densely forested areas, almost all the models have similar performance; however, in dense areas, NDVI- and CHM-based model have similar performance to BASE. Using the DCE metric, we see better results in open and in canopy edges in NDVI and CHM based models than in the rest of the models. DEM derived models again are the lowest performing models in open and canopy edges. Under canopy, the BASE model is slightly better than NDVI but is in the vicinity of the CHM and DEM only model. Surprisingly, in the under canopy, slope-aspect-northness model has comparable performance to other top performing models. Using a height-based threshold, all the models were comparable in performance in forested areas.

| Canopy Metrics | CD | | | DCE | | | CH | |
| --- | --- | --- | --- | --- | --- | --- | --- | --- |
| Model | Sparse | Medium | Dense | Open | CE | UC | OA | FA |
| Base + DEM | 0.87 | 0.98 | 0.98 | 0.70 | 0.79 | 0.92 | 0.87 | 0.98 |
| BASE [1] + slope + aspect + northness | 0.85 | 0.97 | 0.98 | 0.64 | 0.73 | 0.93 | 0.85 | 0.98 |
| BASE [1] + DEM (Elevation) + slope + aspect + northness | 0.79 | 0.94 | 0.96 | 0.51 | 0.65 | 0.88 | 0.79 | 0.95 |
| BASE [1] + NDVI | **0.90** | **0.98** | **0.99** | **0.79** | **0.84** | **0.94** | **0.90** | **0.98** |
| BASE [1] + CHM (Canopy height) | 0.88 | 0.98 | 0.99 | 0.75 | 0.80 | 0.93 | 0.87 | 0.98 |
| BASE [1] | 0.85 | 0.98 | 0.99 | 0.69 | 0.78 | 0.95 | 0.85 | 0.98 |

[1] Refers to Visible + NIR bands from Tuolumne basin only.

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
