# Peer review of "High-Resolution Snow-Covered Area Mapping in Forested Mountain Ecosystems Using PlanetScope Imagery"

_remotesensing, doi:10.3390/rs14143409_

Round 1

Reviewer 1 Report

The goal of this article is  to Improve high resolution (meter-scale) mapping of snow-covered areas in complex and forested terrain,using high-resolution PlanetScope imagery  and CNN algorithm,based on the developed model (Base model, Canistra et al., 2021).This work is meaningful research. However, there are obvious deficiencies in the research scheme. More work should be done to improve the conclusion of the article. The main comments are as follows.

1.What is the basis for adopting several combinations of features (predictor)? I can well understand the physical significance of the predictor combinations of “BASE + CHM (Canopy)” and “BASE + DEM”, because canopy and DEM information are not considered in the BASE model.

But why do you consider BASE + NDVI and not BASE + NDSI? I think NDVI has also been included in the 4-bands (Visible and NIR) of BASE.

In addition, why not consider "BASE + DEM +CHM", "BASE + NDVI+CHM", etc.

When using CNN method to solve the research objectives of this paper, feature selection is very important, and different features selected will lead to different conclusions. Therefore, the selection of features should be based on a comprehensive consideration. This is the biggest defect of this paper and should be improved.

2. PlanetScope imagery is high resolution data. So, the 30m SRTM DEM should be used instead of the 90 m DEM. In line 491,“the SRTM based model had higher precision than the 3m model in 491 the Gunnison evaluation”,This is unbelievable and even pointless.

3. It is suggested to add the distribution map of DEM, CHM, NDVI, NDSI and other features, in Fig 8 and 9

Reviewer 2 Report

Dear authors,

       This paper augmented the Cannistra et al. 2021 convolutional neural networks snow cover model by using additional input data including vegetation metrics (Normalized Difference Vegetation Index) and DEM-derived metrics (elevation, slope and aspect) to improve SCA mapping in forested and open terrain, evaluated the model performan-ce at two geographically diverse sites. This work is interesting and worth publishing. However, there are certain issues with your methods and results should be resolved.

Below are my detailed major comments.

1.Introduction: (Lines 141-146): very awkward sentence, please reorganize.

2.Study Area: (Lines150-167): You mentioned three sites are dissimilar in elevation, climate, and differing forest cover and climatic zones, but only the elevation and forest cover are introduced in the detailed introduction later, it is recommended to be more detailed. In Figure 1 (3), You need to adjust the size and latitude and longitude interval appropriately.

  1. Methods: (Title): This chapter contains the data and methods used, and the title should be written as “Data and Methods”.

(3.1. Data products) (Lines184-189): DEM and canopy height are suggested to be introduced separately. Why you used this data should be presented in this section. The present description is unclear.

(3.2. and 3.3): “we added additional layers that covered…”. I think section 3.2 is a part of model augmentation.

Figure 3 should include more information. Suggestion: 1)it is better to highlight the  augmentation content to clearly show the difference of the method in this study from the original method; 2) all input data are suggested listing out in figure 3.

If I don’t see figure 3, I will not know Lidar data is used for snow depth derivation.

(3.4. Model evaluation in forested areas) : figure 4 : use color figure.

3.4.1, 3.4.2 and 3.4.3 are suggested to be merged together.

(3.5. Model performance metrics) What is the difference between "True Positives" and “TP”, and why should they be expressed separately in these formulas?(“False Positives” and“FP”,etc.) it is suggested to add confusion matrix to make the formula clearer?

(3.6. Evaluation of Lidar derived SCA threshold) : what data did you use to validate Lidar-derived snow depth? Why need a threshold. I am very confused about this section.

Content of section 3.7 should be moved to Discussion.

It is not necessary to present evaluation in section method.

Section 4 is not result but evaluation

Section 4.4 should be content in discussion

4.Result: (4.1 Overall model performance): In Table1,  in overall model performance, metrics were calculated across 19 scenes in Gunnison, and 2 scenes in Engadine, but the amount of data used in Tuolumne is not mentioned(training/validation), and do they need to be shown in Figure 2?

In Figure 6 and in Figure 7, why only Gunnison and Engadine overall performance was presented. in addition, the green is not obvious, especially in black contrasts. Moreover, black is a bit depressing.

 (Line 418-419 and 5.2. Effect of DEM (Elevation) and its derived attributes) the BASE+NDVI model performs best, but the model with DEM and derived attributes is able to get correct snowfall predictions on ridges and valleys:  have you considered whether combining BASE, NDV and DEM might be better?

(4.2. Evaluation of SCA over open and forested areas using canopy metrics at Gunnison): The previous results show that the overall model performance is different in different regions, why only the evaluation of SCA over open and forested areas using canopy metrics at Gunnison is verified, please supplement the verification results in other places, as mentioned in your study area (Study Area Line115-159) and data presentation (3.1. Data products)

(4.3. Evaluation of Lidar derived SCA threshold): This section needs to be adjusted and placed in the method or before 4.1, and the article does not mention the final threshold used.

Figures: make sure text in figures clear

Section 5: shorten the length of the first two paragraph.

Appendix B: In the title of Table B.2, you need to amend "19 Planet Scene IDs" to "2 Planet Scene IDs".

Reviewer 3 Report

This paper is interesting and eligible to be published in Remote Sensing. However, some major revisions must be needed illustrated below. 

1. Please, explain the reasons for using the SRTM DEM with 90 m resolution. SRTM DEM with 30 m can be better for this research. It might be downloaded through EarthExplorer Website. 

2. Please, explain the reasons for using NDVI for this research. There are other vegetation indices. Otherwise, could you compare the various vegetation indices for this research?

3. Usually, SAR imagery such as Sentinel-1 has been used for the snow covered area mapping tasks. Why did you use the optical imagery for this research. You explain the pros and cons between SAR and optical imagery for this research.

Round 2

Reviewer 1 Report

The author makes no effort to address the first comment raised (The most important comment).    I think only when this problem is solved can the scientific value of this manuscript be highlighted.  The authors can compare the accuracy of many more different feature combinations.

And you said " one way to go about it that we used in our study is to use judicious combination of predictors based on previous findings."    But the reader can't see it.    For example, based on my research experience, I think maybe "BASE + DEM +CHM" or "BASE + DEM +NDVI" will be better than the feature combination method in the paper.

1.What is the basis for adopting several combinations of features (predictor)? I can well understand the physical significance of the predictor combinations of “BASE + CHM (Canopy)” and “BASE + DEM”, because canopy and DEM information are not considered in the BASE model.

But why do you consider BASE + NDVI and not BASE + NDSI? I think NDVI has also been included in the 4-bands (Visible and NIR) of BASE.

In addition, why not consider "BASE + DEM +CHM", "BASE + NDVI+CHM", etc.

When using CNN method to solve the research objectives of this paper, feature selection is very important, and different features selected will lead to different conclusions. Therefore, the selection of features should be based on a comprehensive consideration. This is the biggest defect of this paper and should be improved.

Reviewer 2 Report

This study augmented DNN through adding NDVI and topography factors to derive snow cover in forested mountain areas, and analyzed the derive results and limitations. This study adopted planet data which have high spatial and temporal resolution. However, some issues should be resolved before acceptance.

LiDAR-derived snow depth was considered as the true snow cover data. But it was not presented in data description.

There was no detail introduction of Planet data.

What’ the full name of ASO?

Section 3.4 model evaluation in forested areas

This section is confusing. The purpose is to evaluate the inputs or the snow cover resulted from adopting different inputs?

What data did you used for evaluation? Modeled results?

Section 3.6 seems to introduce input data DEM.

Suggest authors to divided section 3 into section 3.1 data and section 3.2 method

If the threshold to derive snow cover from Lidar snow depth introduce error for the training?

Are you sure LiDAR is more accurate than optical remote sensing with high spatial resolution (0.7-3m) to derive snow cover?

In the derivation scheme, DEM and NDVI were not combined to derive snow depth? From table 1, DEM and NDVI have biggest influence on the results, do you consider combine these two together?

Although the paper described the classification of forest, eg. The distance to canopy edge, the canopy density, it was unclear how to parameterize them in model.

Reviewer 3 Report

In Figure 11, check 90% SRTM. please replace 90m with 30m.
